# Considering Forward Electricity Prices for a Hydro Power Plant Risk Analysis in the Brazilian Electricity Market

Arthur Lauro [1], Daniel Kitamura [1], Waleska Lima [1], Bruno Dias [1] and Tiago Soares [2,*]

[1] Electrical Energy Department, Federal University of Juiz de Fora, UFJF, Juiz de Fora 36036-330, Brazil
[2] Center for Power and Energy Systems, Institute for Systems and Computer Engineering, Technology and Science, 4200-465 Porto, Portugal
* Correspondence: tiago.a.soares@inesctec.pt

**Abstract:** The Brazilian Power System is mainly composed of renewable generation from hydroelectric and wind. Hence, spot and forward electricity prices tend to represent the inherently stochastic nature of these resources, while risk management is a measure taken by agents, especially hydro power plants (HPPs) to hedge against deep financial losses. A HPP goal is to maximize its profit considering uncertainties in forward electricity prices, spot prices, and generation scaling factor (GSF) for years ahead. Therefore, the objective of this work is to simulate the real decision-making process of a HPP, where they need to have a perspective of the forward market and future spot price assessment to negotiate forward electricity contracts. To do so, the present work models the uncertainty in electricity forward prices via two-stage stochastic programming, assessing the benefits of the stochastic solution in comparison to the deterministic one. In addition, different risk aversion levels are assessed using conditional value at risk (CVaR). An important conclusion is that the results show that the greater the HPP risk aversion is, the greater the energy selling via electricity forward contracts. Moreover, the proposed model has benefits in comparison to a deterministic approach.

**Keywords:** decision making under uncertainty; electricity forward price; stochastic programming; risk management; renewable energy sources

## 1. Introduction

### 1.1. Motivation and Background

In electricity markets, a player must deal with different types of uncertainties, such as spot prices, forward contract prices, regulation, and resource availability. Therefore, players must perform an exhaustive financial analysis to accomplish their goal, which usually involves maximizing profit while performing risk management. The latter can be done via negotiating forward contracts to hedge spot prices (no capacity investment) or via portfolio diversification with different assets (capacity investment) [1].

The financial risk analysis of a HPP in the Brazilian electricity market considers the mentioned elements while also appraising the uncertainty in inflows to reservoirs and HPP's generation levels. Both aspects are uncontrollable since the HPP's generation is determined via a tight pool centralized chain of models [2]. Then, it is important to thoroughly contemplate the decision-making process and HPP's risk aversion level in favour of maximizing HPP's objectives and hedging against deep financial losses.

Therefore, this work presents a model that simulates the real decision-making process of a HPP, where they need to assess the forward contract prices and spot price development to negotiate electricity contracts at the right time.

### 1.2. Literature Review

Usually, risk analysis regarding the electricity market takes into account the retail agent. In [3], the authors propose a financial hedge method for a retailer with fixed price contracts.

With the exposure of its portfolio to hourly load variations and real-time market prices, the authors propose a risk-averse stochastic programming model with hourly periods for hedging. The Value at Risk (VaR) and Conditional Value at Risk (CVaR) metrics are used in the model. In [4], the authors aim to maximize the profit of a retailer with Renewable Energy Source (RES) generation. For that, a risk-averse stochastic programming model is proposed to define the bidding in the day-ahead market, as well as the participation of its consumers in the demand response (DR) market. The results showed that participation in the DR market raises the retail agent's profit considerably. In addition, the authors in [5] propose a risk-averse two-stage stochastic programming model with the CVaR that assists retailers in dividing their portfolio between forward contracts and bids to the day-ahead market. The authors also analysed how retailers could sell energy to their consumers: flat tariff, time-of-use (TOU) tariff or real-time price tariffs. The results showed that the best option is to offer the type of contract according to the load profile and uncertainty of each consumer. Note that only [5] considers medium and long-term contracts, but not its volatility.

With respect to agents with RES generation, the authors in [6] showed that an agent with a wind generator and energy storage has higher profits than an agent with only one of them. This is explained by the fact that this agent can offer the market a generation with less uncertainty and even with a more desirable profile in the face of the projection of hourly prices, and also has the option to take part in the reserve market. The authors used a risk-averse stochastic programming model with the CVaR metric. In [7], the authors propose a risk-constrained two-stage stochastic optimization model for the decision-making of a RES aggregator with clean energy participating in the day-ahead market. The model can be useful for planning different sources, allowing for their further integration. The authors in [8] applied a downside risk constraints method to define the operation of a pumped-storage HPP in the electricity market. Note that none of these works ponder about mitigating financial risk through forward contracts.

In addition to the risk analysis of different agents in the electricity market, another important aspect is the expansion of generation considering these risks. In [1], the authors use bilevel programming to assist the decision-making of a generating agent that has to decide its positioning in the future market, its exposure in the spot market, and whether to invest in a new generating unit. The results showed that one must invest in a new generating unit and that the positioning in the forward market depends on the arbitrage opportunity, which is limited due to the competitiveness in both markets. Regarding the electricity market in Brazil, the authors in [9] address the portfolio optimization problem through generation expansion using one or more RESs and the diversification of contracts between the free and regulated markets. The proposed model also uses the CVaR and the results showed that a portfolio composed of complementary sources leads to higher profits, and risk-neutral agents seek to compose the portfolio with more free market contracts, while risk-averse agents tend to do so with more regulated market contracts. The models proposed in both works assume that the agent has only one moment to decide on the investment, not contemplating forward price volatility.

Regarding Firm Energy Certificates (FECs) in the Brazilian electricity market, HPPs can choose on a monthly basis how they allocate their electricity contracts. In [10], the authors propose a stochastic, risk-averse and game theory-based methodology for deciding the next year's monthly allocation of FEC. The proposed model takes into account the uncertainty in demand, electricity prices, GSF and other players' behaviours and preferences to find the optimal FEC monthly allocation strategy for a HPP. Note that this work proposes optimizing FEC's monthly allocation and do not deal with electricity prices.

### 1.3. Main Contributions

Considering a free electricity market, HPPs are able to negotiate forward contracts years ahead or to be exposed to spot prices. Thus, this work proposes to model the real decision-making process of a HPP, in which it deals with uncertainties in forward

contract prices, future spot prices, and the GSF. This is an important feature of the decision-making process since HPPs can have the perspective of market forward price evaluation to negotiate forward electricity contracts at the right time. Appraising this feature is the main contribution of this work.

To do so, this work models uncertainty in forward contract prices in a two-stage stochastic problem while also handling uncertainty in future spot prices and GSFs. The proposed model will be evaluated according to the benefit of the stochastic forward contract prices formulation compared to a deterministic one.

In addition, given the nature of the problem, this work will also assess different levels of risk aversion for a HPP using the CVaR approach.

### 1.4. Paper Organization

The rest of this paper is organized as follows: Section 2 presents an overview of the Brazilian market organization. Section 3 is a review of stochastic programming and risk modelling mathematical formulation. Section 4 formulates the proposed model by applying stochastic programming and risk modelling to the HPP problem. Section 5 provides an analysis of the proposed model based on real data from the Brazilian electricity market. Section 6 delivers the most important conclusions of this work.

## 2. Brazilian Electricity Market Overview

The Brazilian power system is a large-scale hydro-thermo-wind system centrally dispatched by the independent system operator (ISO). The Operador Nacional do Sistema Elétrico (ONS), the Brazilian ISO, is responsible for optimizing the National Interconnected System dispatch to minimize the operating cost while meeting the load demand. To do so, a chain of optimization models for different planning horizons is required [2], namely:

- Long term: due to the large participation of HPPs in the Brazilian electricity matrix, which accounts for more than 60% of the total installed capacity, uncertainty about future hydrological conditions makes it necessary to consider the dispatch of the system five years ahead. For this purpose, the optimization model is used on a monthly basis that emphasizes the stochastic inflows and simplifies the NIS representation, such as considering the aggregation of individual reservoirs into equivalent reservoirs. This horizon usually sets forward prices for the years ahead, A+1 until A+5, where A is the current year.
- Short term: aiming at planning the system a few months ahead, it is necessary to simulate the dispatch on a weekly basis and with individual reservoirs. In this horizon, the forward prices are set for the months ahead.
- Hourly Basis: aiming at setting the day-ahead operation. It is necessary to model the system in as much detail as possible, with transmission line flow limits and generation unit constraints. For this horizon, the inflow is known, and the model provides the day-ahead hourly prices.

The Lagrange multipliers of the load demand-supply equation define the spot price for each area and are named Marginal Operating Cost (CMO). The CMO, limited by regulatory limits, is named at Settlement Price for the Differences (PLD) and is used to evaluate the Short-Term Market financial surplus. The PLD is defined on an hourly basis and for each Brazilian area: Southeast, South, Northeast, and North. These day-ahead prices are used to financially settle the Brazilian electricity market [11].

### 2.1. Brazilian Electricity Regulatory Environment

Brazilian electricity commercialization can take place in the Free Market (ACL), which has free negotiation between its counterparts, or in the Regulated Market (ACR), through electricity Commercialization Contracts (CCEAR) firmed by auctions held by the Chamber of Commercialization of Electric Energy (CCEE) [11]. Therefore, the electricity price in the ACR is defined from energy auctions settled by the Brazilian National

Agency of Electric Energy (ANEEL), while in the ACL, this price is negotiated considering counterpart expectations.

### 2.2. The Settlement Price and the Generation Scaling Factor

The PLD is obtained by computational models in a central dispatch model. Since the Brazilian electricity system has a matrix composed mainly of hydro power plants, the optimization model has a stochastic characteristic that depends on future inflows. In the face of the inherent uncertainties of the inflows and RES generation, it is possible to observe high volatility in PLD, as indicated in Figure 1.

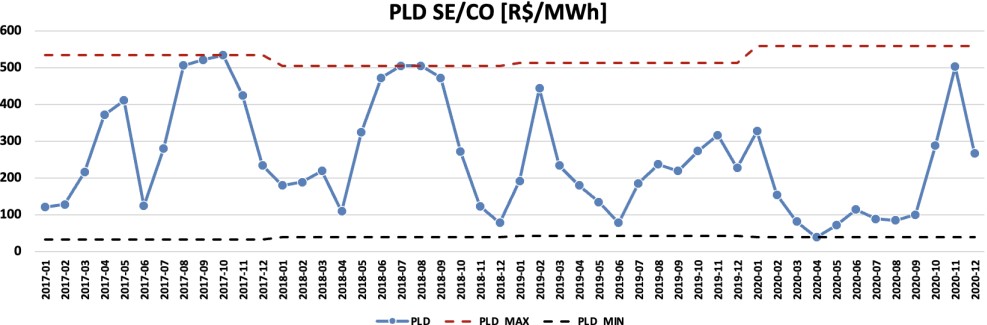

**Figure 1.** Historical Southeast PLD.

Since Brazil has continental dimensions and presents different inflows seasonality for each region, these regions are connected by transmission lines to form the NIS. Therefore, the ONS is responsible for operating the system, optimizing the hydraulic source, while taking advantage of some region surplus to supply other regions. Considering this complementary energy between regions, the Energy Reallocation Mechanism (MRE) was created [12]. The MRE is responsible for sharing the total hydraulic energy generated among the HPPs through a financial instrument. The agents that overshoots their Firm Energy Certificate (FEC) transfer their surplus to the ones that generated below their FEC. The HPP's FEC is calculated considering its capacity to contribute to a critical load considering a critical inflow period. The sum of all MRE agents' FECs is defined as MRE Firm Energy Certificate (MRE-FEC).

In addition, the GSF is evaluated on a monthly basis and is defined as the ratio between the MRE hydraulic generation and the MRE-FEC. This factor represents the NIS hydraulic generation performance. In recent years, the GSF has frequently performed below 1 due to severe hydrological conditions (droughts) nationwide. The monthly GSF values are depicted in Figure 2.

For further discussion about the financial impacts on HPPs caused by the ongoing severe drought in Brazil and the current state of the MRE and FEC, interested readers are directed to [12,13].

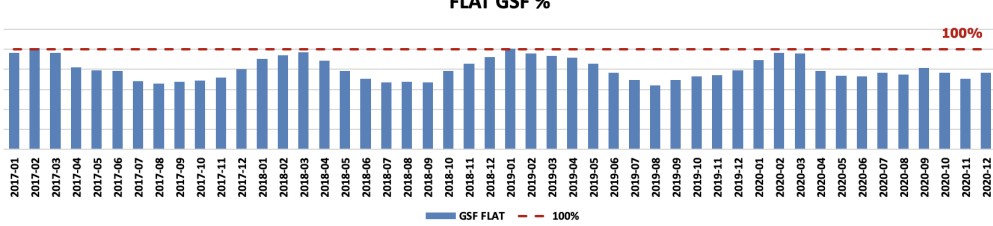

**Figure 2.** Historical Flat GSF.

### 2.3. Forward Electricity Prices

Long-term contracts of different supply horizons essentially protect agents against PLD uncertainty. The forward electricity price negotiated between the ACL counterparts

is calculated based on price expectations and defines a forward curve. This curve sets a reference for the electricity price according to the product maturity. There are different approaches for calculating forward curves in the literature, as described in [14,15].

Price uncertainty affects both short- and long-term contracts. Although the volatility tends to be higher in the short-term, long-term contracts also present uncertainties due to many variables in the market, such as regulatory changes. Thus, this work aims to consider the uncertainty of the forward curve and the GSF expectations to obtain the long-term optimal contract level for a HPP.

Another important aspect of forward price formation in Brazil is that the HPP is procured after a concession auction based on the lowest price of energy sold to consumers. Then, a quota of the HPP's FEC is reserved for the regulated consumers (ACR) and also guarantees a future cash flow increasing the feasibility of the investment [16]. Hence, the Levelized cost of electricity (LCOE) is not considered in this paper as the LCOE is more relevant in project than in energy trading decision-making.

## 3. Stochastic Programming and Risk Modeling

### 3.1. Two-Stage Stochastic Programming

A decision-making problem under uncertainty is a process that models a timely sequence of optimal decisions made under uncertainty. Two-stage stochastic programming techniques are applied to solve this problem, whether the stages model when the decision-maker has to make a decision and when the uncertainty is taken into account [17,18]. The two-stage decisions can be defined as follows:

1.  First-stage or here-and-now: models the decision made before the realization of uncertainty, i.e., it does not depend on each realization of the stochastic process.
2.  Second-stage or wait-and-see: models the decisions made after the realization of the uncertainty. It models the corrective actions and consequences of the first-stage decision according to the realization of the stochastic process.

The mathematical modeling of the two-stage stochastic problem is defined as follows: [7,17,18]:

$$Max \ c^T x + \varepsilon\{Q(w)\} \tag{1}$$

subject to:

$$Ax = b \tag{2}$$

$$x \in X \tag{3}$$

where:

$$Q(w) = \left\{ Max \ q(w)^T y(w) \right. \tag{4}$$

$$subject \ to: \ T(w)x + W(w)y(w) = h(w) \tag{5}$$

$$\left. y(w) \in Y \ \forall w \in \Omega \right\} \tag{6}$$

where $x$ are the first-stage variables and $y(w)$ are the second-stage variables. The stochastic process is represented via scenarios by $w$. $c$, $q(w)$, $b$, $h(w)$, $A$, $T(w)$ and $W(w)$ are known vectors and matrices. Equations (1)–(3) represent the first-stage problem, while Equations (4)–(6) model the second-stage problem, which is a function of the decision made on the first-stage problem. Equations (1)–(6) can be expressed as follows [17,18]:

$$Max \ c^T x + \sum_{w \in \Omega} \pi(w)q(w)^T y(w) \tag{7}$$

subject to:

$$Ax = b \tag{8}$$

$$T(w)x + W(w)y(w) = h(w), \ \forall w \in \Omega \tag{9}$$

$$x \in X, \ y(w) \in Y, \ \forall \omega \in \Omega \tag{10}$$

where $\pi(w)$ is the probability of scenario $w$.

### 3.2. Risk Modeling

Risk management is very important when dealing with uncertainty in an investment portfolio. Risk can also be modeled in an optimization problem, and a well-known metric is the CVaR, which is a coherent risk measure [19]. This metric quantifies the risk of an investment or decision by pondering the expected return in all scenarios and the expected return of a quantile of the worst scenarios. Namely, given $\alpha \in (0, 1)$, CVaR can be expressed as the expected value of a quantile-$(1 - \alpha)$ of a distribution [17]. The mathematical model is represented as follows and is based on [7,17]:

$$Max \ \eta - \frac{1}{1 - \alpha} \cdot \sum_{w=1}^{W} \pi_w s_w \tag{11}$$

subject to:

$$\eta - (c^T x + q(w)^T y(w)) \leq s_w \ \forall \omega \in \Omega \tag{12}$$

$$s_w \geq 0 \ \forall \omega \in \Omega \tag{13}$$

where $\alpha$ is a confidence level, and $\eta$ and $s_w$ are auxiliary variables.

CVaR can be introduced in (7)–(10) as follows:

$$Max \ \ (1 - \beta) \cdot \left( c^T x + \sum_{w \in \Omega} \pi(w) q(w)^T y(w) \right) + \beta \cdot \left( \eta - \frac{1}{1 - \alpha} \cdot \sum_{w=1}^{W} \pi_w s_w \right) \tag{14}$$

subject to: (9), (10), (12), and (13). $\beta \in (0,1)$ is the risk aversion parameter. For example, $\beta = 0$ means that the decision-maker is neutral to risk, and $\beta = 1$ expresses maximum risk aversion.

## 4. Proposed Model

### 4.1. Decision-Making Process of a Hydro Power Plant

In the Brazilian electricity market, the optimal portfolio and contract management of a HPP in the long term, i.e., one year (A+1) or five years (A+5) ahead, depends on market expectations of future PLD, on future GSF and on the evolution of those forward electricity contract prices. The latter is modeled considering the uncertainties of forward contracts.

Figure 3 shows the flowchart of the proposed decision-making methodology and Figure 4 details the decision-making process over time. One can note that in the first stage (n = 1), the HPP has full knowledge of A+1 (one year ahead) electricity forward contract price $\lambda_{n=1}$. Similarly, the HPP has a set of scenarios of PLD and GSF for each month t of year A+1. However, the HPP does not need to make the portfolio decision in stage n = 1 because the A+1 electricity forward contract price can (and will) change while some of the uncertainty vanishes and the distribution of one year ahead PLD and GSF evolves.

Then, the decision-making process of years ahead electricity forward contract price uncertainty can be formulated as follows:

1.  Define $n = 1, \dots, N$ electricity forward contract negotiation stages
2.  Define $f = 1, \dots, F$ electricity forward contract price scenarios
3.  Define $w = 1, \dots, W$ scenarios of PLD and GSF for each electricity forward contract price scenario f
4.  evaluate the expected profit considering that the decision in n = 1 is the same for all scenarios f (here-and-now decision)

In addition, the HPP is not able to negotiate more energy than their FEC.

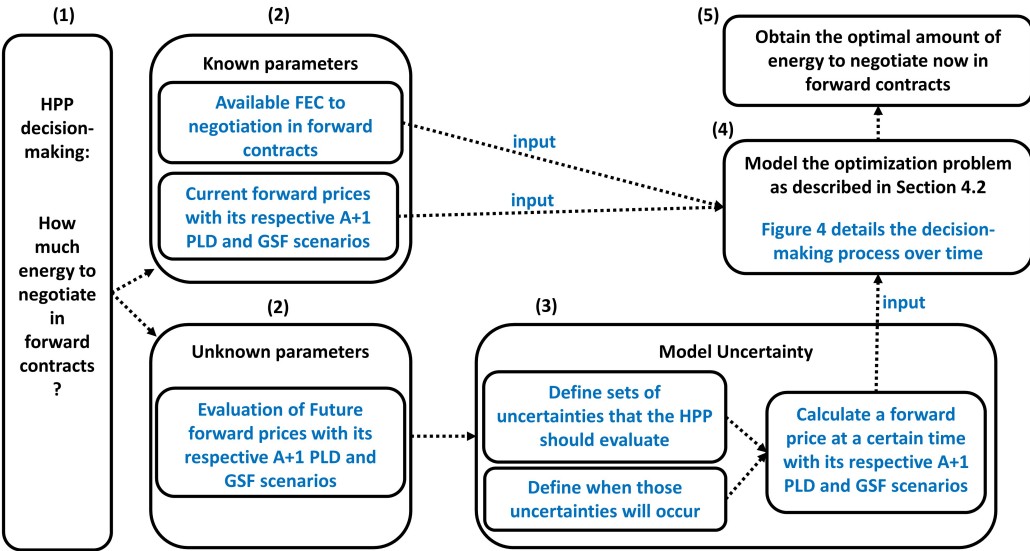

**Figure 3.** Proposed decision-making methodology flow chart.

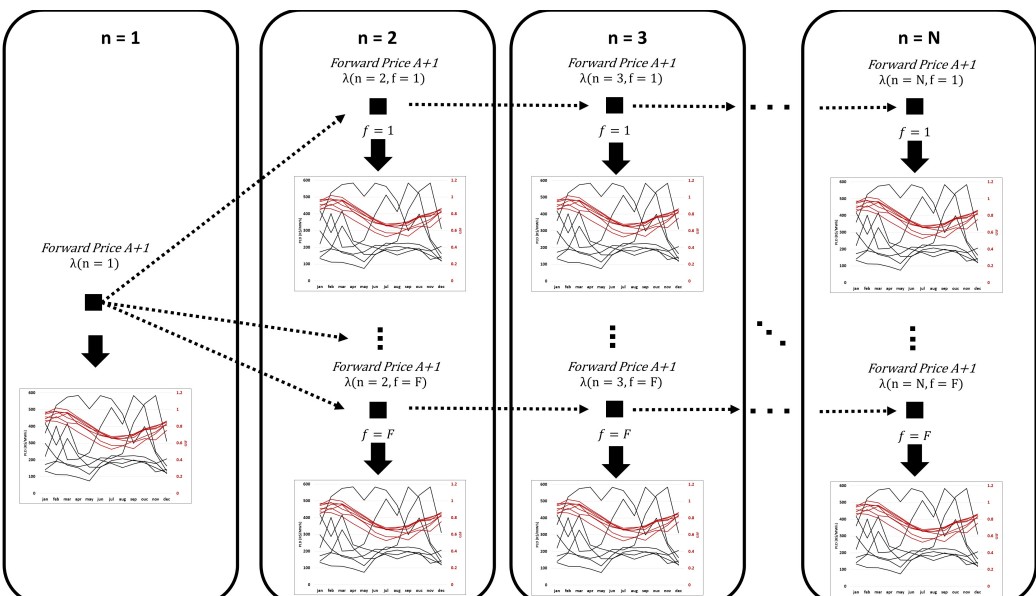

**Figure 4.** Decision-making process flow chart of a HPP participation in the Brazilian electricity market.

## 4.2. Mathematical Modelling

In this subsection, the full modelling of the HPP participating in the Brazilian electricity market under uncertainty and risk management (based on [17]) can be written as:

$$
Max (1 - \beta) \cdot \left( \sum_{f=1}^{F} \pi_f \cdot \sum_{n=1}^{N} \sum_{t=1}^{T} \lambda_{f,n} x_{f,n} + \right.
$$

$$
\left. + \sum_{f=1}^{F} \pi_f \cdot \sum_{w=1}^{W} \pi'_{f,w} \cdot \sum_{t=1}^{T} y_{f,w,t} \right) +
$$

$$
+ \beta \cdot \left( \eta - \frac{1}{1-\alpha} \cdot \sum_{f=1}^{F} \sum_{w=1}^{W} \pi_f \pi'_{f,w} s_{f,w} \right) \quad (15)
$$

subject to:

$$y_{f,w,t} = PLD_{f,w,t} \cdot \left(GSF_{f,w,t} \cdot FEC - \sum_{n=1}^{N} x_{f,n}\right)$$

$$f = 1, \ldots, F, \ t = 1, \ldots, T, \ w = 1, \ldots, W \quad (16)$$

$$x_{f=1,n=1} = x_{f=2,n=1} = \ldots = x_{f=F,n=1} \quad (17)$$

$$0 \leq \sum_{n=1}^{N} x_{f,n} \leq FEC, \ f = 1, \ldots, F \quad (18)$$

$$\eta - \left(\sum_{n=1}^{N} \sum_{t=1}^{T} \lambda_{f,n} x_{f,n} + \sum_{t=1}^{T} y_{f,w,t}\right) \leq s_{f,w},$$

$$f = 1, \ldots, F, \ w = 1, \ldots, W \quad (19)$$

$$s_{f,w} \geq 0, f = 1, \ldots, F, \ w = 1, \ldots, W \quad (20)$$

where:

- $\pi_f$ is the probability of electricity forward contract price scenario $f = 1, \ldots, F$
- $\pi'_{f,w}$ is the probability of PLD scenario $w = 1, \ldots, W$ of electricity forward contract price scenario $f = 1, \ldots, F$
- $\lambda_{f,n}$ is the electricity forward contract price scenario $f = 1, \ldots, F$ on stage $n = 1, \ldots, N$
- $x_{f,n}$ is the amount of electricity negotiated on forward contract price scenario $f = 1, \ldots, F$ on stage $n = 1, \ldots, N$
- $y_{f,w,t}$ is the spot market revenue on intra-stage $t = 1, \ldots, T$ of PLD scenario $w = 1, \ldots, W$ of electricity forward contract price scenario $f = 1, \ldots, F$
- $PLD_{f,w,t}$ is the spot price realization of intra-stage $t = 1, \ldots, T$ of PLD scenario $w = 1, \ldots, W$ of electricity forward contract price scenario $f = 1, \ldots, F$
- $GSF_{f,w,t}$ is the GSF of intra-stage $t = 1, \ldots, T$ of PLD scenario $w = 1, \ldots, W$ of electricity forward contract price scenario $f = 1, \ldots, F$
- FEC is the hydro power plant's Firm Energy Certificate
- $\eta$ and $s_{f,w}$ are CVaR auxiliary variables

The objective function (15) expresses the profit on electricity forward contract negotiation and on spot market revenues (SMR). The SMR is defined in Equation (16) as the spot price valuation of the difference between the total negotiated future electricity contracts and the product of that month's GSF and its FEC. Equation (17) assures that the first-stage decision is the same for all forward contract price scenarios. Next, Equation (18) imposes that the HPP is not able to sell more energy on forward contracts than their FEC. Equations (19) and (20) are due to CVaR modelling.

Additionally, one can consider $GSF_{f,w,t} = 1.0$ (to all stages and scenarios) for a generalization to other electricity markets that do not have a GSF mechanism.

## 5. Case Study

This section presents a test case of decision-making under uncertainty of a HPP, which can decide one year ahead (A+1) when and how much energy to negotiate on electricity forward contracts and how much to be exposed on spot market contracts. Therefore, this case study defines the HPP portfolio for one year ahead (A+1).

It is considered that the HPP FEC is 100 MW flat all year round. Although we have presented a case study with 100 MW, the results are insensitive to the FEC's actual value. Thus, we present the decision in the percentage of FEC rather than its amount in MW.

For each HPP portfolio decision stage, the spot market and GSF scenario are monthly while the forward contract is annual. The HPP has two stages to decide "when" and

"how much" energy to negotiate on forward contracts. It can be highlighted that only the first-stage decision will be taken into account, as the HPP will be able to reassess its next stages' decisions in the near future with more information and the realization of some uncertainties. This case study also assesses the benefit of the stochastic modelling of electricity forward contract prices via the Value of the Stochastic Solution (VSS) metric. In addition, the efficient frontier of the risk aversion parameterization will be displayed.

The following subsections describe the stochastic process of PLD, GSF and electricity forward contract prices.

### 5.1. PLD and GSF Scenarios

For the one-year-ahead analysis (A+1), the monthly PLD and GSF scenarios are obtained through the assessment of the chain of optimization models. This is a commonly adopted practice in Brazil's electricity market since these models define each NIS generation unit dispatch and determine the spot price.

For this paper, a set of 75, 12-month scenarios of PLD and GSF was generated by the described process, and they will be valid for all electricity forward contract price scenarios. Note that the PLD and GSF scenarios could vary for each forward contract price scenario.

Figure 5 displays in blue the permanence curve of the annual mean value of PLD on the left vertical axis and their respective annual mean value of the GSF on the right vertical axis in red. The mean annual PLD is 207.00 R$/MWh.

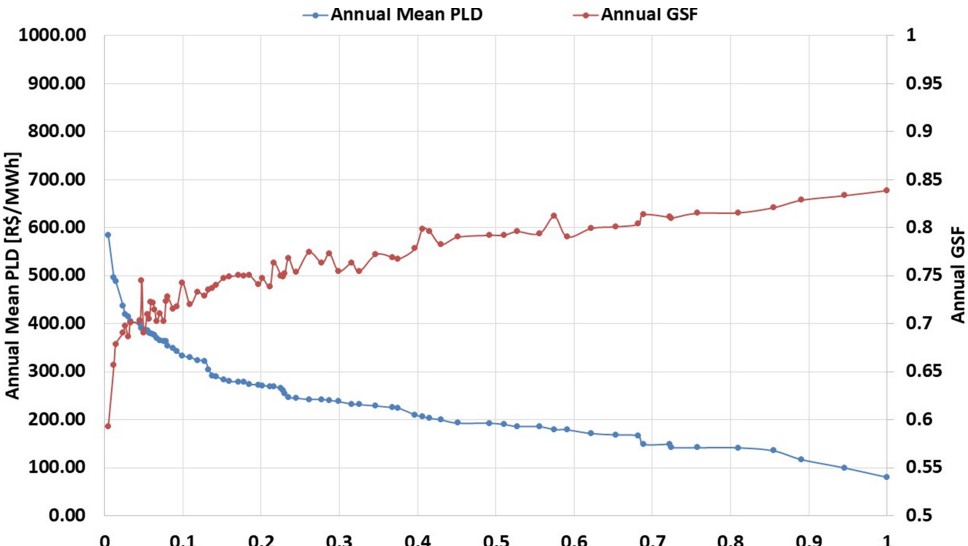

**Figure 5.** Permanence curve of the annual mean value of PLD (blue, left axis) and their respective annual mean value of the GSF (red, right axis).

### 5.2. Forward Prices

As described in the introduction section, there are many methods to determine the electricity forward contract price. As it is not the main goal of this work, we considered that the negotiated electricity forward contract price in the first stage is equal to the mean annual PLD of one year ahead (A+1)-207.00 R$/MWh. For the second stage, we take into consideration a normal distribution with 207.00 R$/MWh as the mean and 20.70 R$/MWh (10%) as the standard deviation. Table 1 displays the electricity forward contract price scenarios.

**Table 1.** Forward contract price scenarios.

|  | Scenario 1 | Scenario 2 | Scenario 3 | Scenario 4 | Scenario 5 |
|---|---|---|---|---|---|
| Forward Contract Price Stage 1 (R$/MWh) | 207.00 | 207.00 | 207.00 | 207.00 | 207.00 |
| Forward Contract Price Stage 2 (R$/MWh) | 165.00 | 186.00 | 207.00 | 228.00 | 249.00 |
| Probability | 0.09070 | 0.23863 | 0.34134 | 0.23863 | 0.09070 |

*5.3. Results*

The proposed model was implemented in GAMS and was evaluated for two cases: (i) electricity forward contract price modeled as a deterministic value, i.e., 207.00 R$/MWh, and (ii) electricity forward contract price modeled as a stochastic value, as shown in Table 1. In both cases, the CVaR methodology was applied for the $(1 - \alpha) = 10\%$ worst scenarios and for different risk aversion levels $(\beta)$.

Figure 6 shows the efficient frontier for the deterministic case. Note that more risk-averse HPP can have a greater CVaR (+92.3%) without having a significant reduction in the expected profit ($-0.2\%$). This result indicates that any degree of risk aversion protects the HPP from the worst scenarios without reducing its expected profit.

For this deterministic case, the optimal energy selling via forward contracts is 60.43% of FEC except for the risk-neutral plan that indicates no energy selling, as indicated in Figure 7. One can note that the amount of non-negotiated FEC is available for future decisions.

Figure 8 shows the efficient frontier for the stochastic case. Notably, the greater the risk aversion level is, the lower the expected profit. This result demonstrates the trade-off between better protection from the worst scenarios and the expected profit.

Also, the optimal energy selling via forward contracts for different risk aversion levels is displayed in Figure 9 as a percentile of its Firm Energy Certificate (FEC). One can note that a less risk aversion HPP prefers to wait on selling electricity in the forward market favoring the higher expected profit whereas a more risk aversion agent tends to sell electricity now for better coverage of losses.

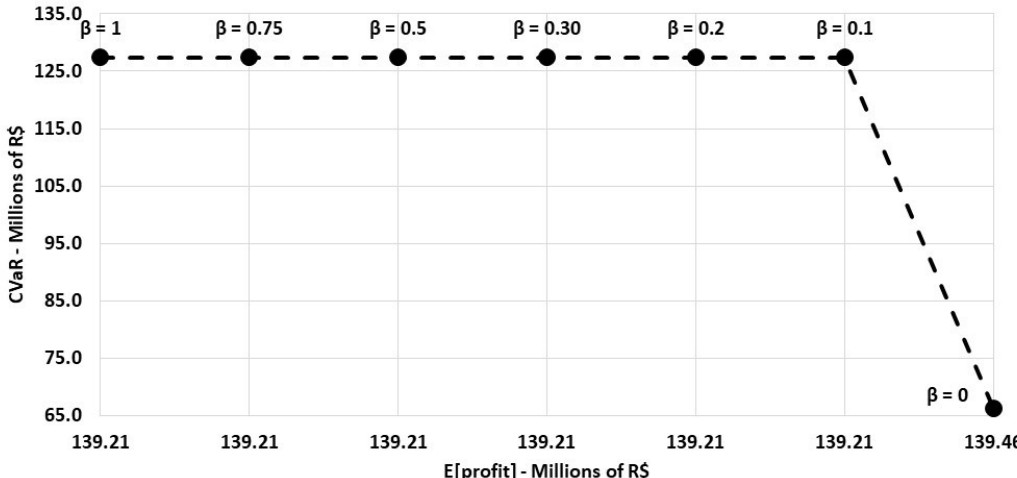

**Figure 6.** Efficient frontier for the deterministic case.

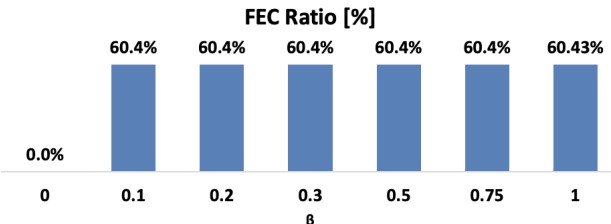

**Figure 7.** Optimal energy selling via forward contracts for different risk aversion levels-deterministic case.

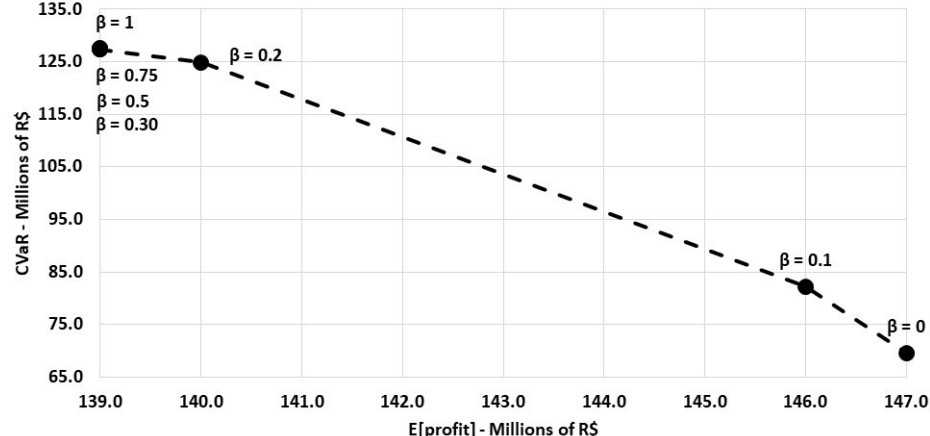

**Figure 8.** Efficient Frontier for the stochastic case.

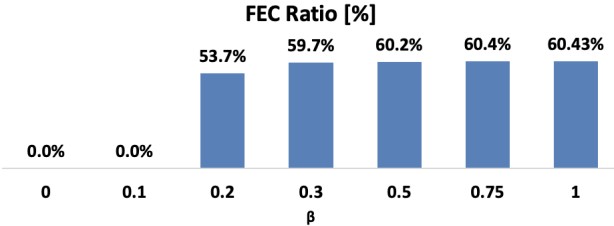

**Figure 9.** Optimal energy selling via forward contracts for different risk aversion levels-stochastic case.

In addition, the benefit of the stochastic modeling of the electricity forward contract prices via the VSS metric was assessed. For a maximization problem, VSS is defined by the difference between the optimal value of the stochastic problem (RP) and the value of the objective function of the stochastic problem while applying the deterministic problem solution (EEV). For further details, see [20].

Table 2 shows the absolute and relative values of VSS for different levels of risk aversion. One can observe that stochastic modeling is more convenient for lower risk aversion levels as indicated by the higher VSS . This analysis highlights the benefits of applying the proposed method.

**Table 2.** Stochastic model benefit evaluation.

|  | RP (R$) | EEV (R$) | VSS (R$) | VSS(%) |
|---|---|---|---|---|
| $\beta = 0$ | $1.47 \times 10^8$ | $1.39 \times 10^8$ | $7.59 \times 10^6$ | 5.2% |
| $\beta = 0.1$ | $1.47 \times 10^8$ | $1.39 \times 10^8$ | $7.79 \times 10^6$ | 5.3% |
| $\beta = 0.2$ | $1.40 \times 10^8$ | $1.39 \times 10^8$ | $1.12 \times 10^6$ | 0.8% |
| $\beta = 0.30$ | $1.40 \times 10^8$ | $1.39 \times 10^8$ | $3.72 \times 10^5$ | 0.3% |
| $\beta = 0.5$ | $1.39 \times 10^8$ | $1.39 \times 10^8$ | $2.92 \times 10^5$ | 0.2% |
| $\beta = 0.75$ | $1.39 \times 10^8$ | $1.39 \times 10^8$ | $1.90 \times 10^5$ | 0.1% |
| $\beta = 1$ | $1.39 \times 10^8$ | $1.39 \times 10^8$ | $1.85 \times 10^5$ | 0.1% |

## 6. Conclusions

This work proposes a methodology that simulates the real decision-making process of a HPP based on the uncertainty of forward contract prices and of years ahead spot prices and GSF. In addition, CVaR was applied to handle the risk in this problem.

The results show that (i) the greater the HPP risk aversion is, the greater the energy selling via electricity forward contracts and that (ii) the proposed model has benefits in comparison to a deterministic approach, highlighted by the VSS metric, especially with low risk-aversion HPP's strategy.

Therefore, the authors concluded that the work reached its goal, which was to model the true decision-making process of a HPP for years ahead aiming at improving portfolio management through stochastic optimization.

Future works will include the decision-making process of a HPP considering hourly spot prices and hydro generation instead of a monthly-basis price, thus evaluating the models under discussion in the Brazilian regulatory studies.

**Author Contributions:** Conceptualization, A.L., D.K., W.L., B.D. and T.S.; Data curation, A.L.; Formal analysis, A.L.; Investigation, A.L.; Methodology, A.L., D.K., W.L. and B.D.; Software, A.L.; Supervision, B.D. and T.S.; Validation, A.L., D.K., W.L. and B.D.; Writing—original draft, A.L., D.K. and W.L.; Writing—review & editing, A.L., B.D. and T.S.; All authors have read and agreed to the published version of the manuscript.

**Funding:** This research was supported in part by Coordenação de Aperfeiçoamento de Pessoal de Nível Superior (CAPES) under Grant 001, Conselho Nacional de Desenvolvimento Científico e Tecnológico (CNPq) under the grants 404068/2020-0, Fundação de Amparo à Pesquisa do Estado de Minas Gerais (FAPEMIG) under grant APQ-03609-17, and Instituto Nacional de Energia Elétrica (INERGE). It is also supported by Norte Portugal Regional Operational Programme (NORTE 2020), under the PORTUGAL 2020 Partnership Agreement, through the European Regional Development Fund (ERDF), within the DECARBONIZE project under agreement NORTE-01-0145-FEDER-000065 and by the Scientific Employment Stimulus Programme from the Fundação para a Ciência e a Tecnologia (FCT) under the agreement 2021.01353.CEECIND.

**Institutional Review Board Statement:** Not applicable.

**Informed Consent Statement:** Not applicable.

**Data Availability Statement:** Not applicable.

**Conflicts of Interest:** The authors declare no conflict of interest.

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
