# Peer review of "Considering Forward Electricity Prices for a Hydro Power Plant Risk Analysis in the Brazilian Electricity Market"

_energies, doi:10.3390/en16031173_

Round 1

Reviewer 1 Report

The manuscript discusses about the “Forward Electricity Prices for a Hydro Power Plant Risk Analysis in the Brazilian Electricity Market.” The author has discussed novel and improved method and compared results with different scenarios 1) by deterministic and 2) stochastic ways.

However, it is highly suggested to add complete flow chart of methodology before publishing article.

Author Response

Dear reviewer,

The answers to your comments are in the attached file.

Reviewer 2 Report

THIS ANALYSIS SHOULD BE CONNECTED TO LCOE (LEVELIZED COS OF ELECTRICITY). THE FORWARD PRICING CAN NOT BE CONSIDERED WITHOUT MENTIONING LCOE. THIS ADDITION WILL INCREASE THE VALUE OF THE PAPER.

Author Response

Dear reviewer,

The answers to your comments can be found in the attached file.

Reviewer 3 Report

This paper focuses on the interesting issue of forward electricity prices, which is of course of fundamental importance for the electricity market. The authors concentrate on the specific case of the Brazilian power market, whereas Brazil heavily relies on hydropower plants for its power generation.

The paper is well written, the methodology is clearly explained, as well as the literature review. Maybe the only limitation is that the results and the conclusions are presented in a pretty concise way, which might not be easy to understand for people who are not working in this precise research field or are not at ease with electricity market dynamics. 

The conclusion is clearly stated but is maybe missing some remarks on the possible generalization and overall significance of the results : one direction could be to go beyond the 100 MW HPP test-case (e.g. what is expected to happen for smaller  or bigger power plants) and to try to generalize to other electricity markets dominanted by hydropower such as Sweden or Canada (also very large countries as Brazil).

Nonetheless, I think that the paper can be accepted in the present form.

Author Response

(The authors gave the same response as above.)

Round 2

Reviewer 1 Report

The suggested corrections has been made.